# Weak Localization of Radiographic Manifestations in Pulmonary Tuberculosis from Chest X-ray: A Systematic Review

**DOI:** 10.3390/s23156781

**Published:** 2023-07-29

**Authors:** Degaga Wolde Feyisa, Yehualashet Megersa Ayano, Taye Girma Debelee, Friedhelm Schwenker

**Affiliations:** 1Ethiopian Artificial Intelligence Institute, Addis Ababa P.O. Box 40782, Ethiopia; degagawolde@gmail.com (D.W.F.); yehualeuven@gmail.com (Y.M.A.); tayegirma@gmail.com (T.G.D.); 2Department of Electrical and Computer Engineering, Addis Ababa Science and Technology University, Addis Ababa P.O. Box 120611, Ethiopia; 3Institute of Neural Information Processing, Ulm University, 89069 Ulm, Germany

**Keywords:** pulmonary tuberculosis, visualization, localization, chest X-ray, weakly supervised

## Abstract

Pulmonary tuberculosis (PTB) is a bacterial infection that affects the lung. PTB remains one of the infectious diseases with the highest global mortalities. Chest radiography is a technique that is often employed in the diagnosis of PTB. Radiologists identify the severity and stage of PTB by inspecting radiographic features in the patient’s chest X-ray (CXR). The most common radiographic features seen on CXRs include cavitation, consolidation, masses, pleural effusion, calcification, and nodules. Identifying these CXR features will help physicians in diagnosing a patient. However, identifying these radiographic features for intricate disorders is challenging, and the accuracy depends on the radiologist’s experience and level of expertise. So, researchers have proposed deep learning (DL) techniques to detect and mark areas of tuberculosis infection in CXRs. DL models have been proposed in the literature because of their inherent capacity to detect diseases and segment the manifestation regions from medical images. However, fully supervised semantic segmentation requires several pixel-by-pixel labeled images. The annotation of such a large amount of data by trained physicians has some challenges. First, the annotation requires a significant amount of time. Second, the cost of hiring trained physicians is expensive. In addition, the subjectivity of medical data poses a difficulty in having standardized annotation. As a result, there is increasing interest in weak localization techniques. Therefore, in this review, we identify methods employed in the weakly supervised segmentation and localization of radiographic manifestations of pulmonary tuberculosis from chest X-rays. First, we identify the most commonly used public chest X-ray datasets for tuberculosis identification. Following that, we discuss the approaches for weakly localizing tuberculosis radiographic manifestations in chest X-rays. The weakly supervised localization of PTB can highlight the region of the chest X-ray image that contributed the most to the DL model’s classification output and help pinpoint the diseased area. Finally, we discuss the limitations and challenges of weakly supervised techniques in localizing TB manifestations regions in chest X-ray images.

## 1. Introduction

### 1.1. Tuberculosis (TB)

The bacteria that cause TB are easily transferred from one infected individual to another since it is contagious. WHO revealed that approximately 10 million individuals had active TB disease in 2019 and that 1.4 million people died from the disease [1]. The 2021 global TB report shows that there was only an 11% decrease in TB incidence, although the goal was to cut TB incidence by 20% between 2015 and 2020 [2]. With only a 9.2% improvement in death rates between 2015 and 2020, the mortality reduction targets, which were set to 35%, were not met [2]. This indicates that TB remained one of the most common causes of death.

The approach of a symptoms inquiry and chest X-ray (CXR) has a higher sensitivity than other approaches, and it mirrors the CXR anomaly in symptomatic people [3]. Most of the time, the clinical decision making to diagnose active tuberculosis is based on the results of a symptoms inquiry questionnaire and chest radiography [4,5]. The sensitivity and specificity of the symptoms inquiry screening questionnaire are 77 and 66 percent, respectively, while they are better in the Purified Protein Derivative (PPD) (89 and 80%, respectively); nevertheless, they are greater in the CXR (86% and 89%, respectively) [6,7]. Symptoms inquiries and CXRs are effective screening methods, particularly in resource-limited settings, but other approaches may also be useful depending on the context.

### 1.2. Radiographic Features of Pulmonary TB

Primary, post-primary, and miliary tuberculosis are terms used to describe different patterns and manifestations of TB [8,9]. Primary tuberculosis occurs when a person is initially infected with Mycobacterium tuberculosis, the bacterium that causes TB. Post-primary tuberculosis, also known as secondary tuberculosis or reactivation tuberculosis, occurs when the bacteria that initially caused the primary infection become dormant in the body but later reactivate due to a weakened immune system. Miliary tuberculosis is a severe and disseminated form of TB where the bacteria spread from the lungs to other organs via the bloodstream, leading to the formation of tiny nodules or lesions resembling millet seeds. The variety of pulmonary tuberculosis (PTB) manifestations depends on whether the infection is primary, post-primary, or miliary. The pathologic site of infection within the lung and the radiographic features differ depending on the patient’s age and infection stage [8,10].

In primary pulmonary tuberculosis, the first areas of infection can be anywhere in the lung and can have any number of non-specific appearances, from patches of consolidation to spots that are too small to be seen. Only 10 to 30% of early TB infections show cavitation, and 70% of children and 90% of adults have radiographic signs of parenchymal infection [11]. The majority of the time, the infection becomes a localized, caseating granuloma (also known as a tuberculoma), and it typically calcifies, becoming a Ghon lesion. The more notable feature is the often right-sided, ipsilateral hilar and contiguous mediastinal (paratracheal) lymphadenopathy, which is particularly common in pediatric cases, Figure 1a,b. Pleural effusions are more frequent in adults, seen in 30–40% of cases, whereas they are only present in 5–10% of pediatric cases [11]. As the host mounts an appropriate immune response, both the pulmonary and nodal disease resolve. When a calcified node and a Ghon lesion are present, the combination is known as a Ranke complex [10,12].

Post-primary infections have a strong propensity for the upper zones. In most cases, the upper lobes’ posterior or lower lobes’ superior segments are where post-primary TB in the lungs typically manifests itself. Patchy consolidation or poorly defined linear and nodular opacities are the usual symptoms of post-primary tuberculosis, and examples are given in Figure 1c,d. Primary infections have a far lower chance of cavitating than post-primary infections. Endobronchial spread along nearby airways is a relatively common finding, resulting in well-defined 2–4 mm nodules. Only about a third of cases have hilar nodal enlargement. Although they are less frequent, lobar consolidation, tuberculoma development, and miliary TB are the other recognized features of post-primary TB. Only 5% of cases of post-primary TB are caused by tuberculomas, which present as a well-defined, spherical mass that is usually found in the upper lobes. They can be up to 4 cm in size and are often solitary. Most frequently, little satellite lesions are observed. Superimposed cavitation may develop in 20–30% of instances [10,12].

Miliary tuberculosis represents the hematogenous dissemination of an uncontrolled tuberculous infection. Miliary tuberculosis is manifested by an even distribution throughout both the left and right lung lobes. It appears as 1–3 mm diameter nodules, which are uniform in size and uniformly distributed, as depicted in Figure 2. When a tracheal infection is isolated, the consequence is irregular circumferential mural thickening, while broncholiths cause calcified material to enter the lumen. Tuberculous involvement of the pleura usually presents as pleural effusion, empyema, or pleural thickening. Diffuse pleural thickening and adhesions with calcification occur in chronic cases. Patients may show a paradoxical reaction on imaging [10,12].

### 1.3. Localization of Pulmonary Tuberculosis in a Chest X-ray

The term “localization” refers to recognizing and pinpointing the disease’s pathology on the chest X-ray. By examining the images, radiologists can identify the precise radiographic characteristics of TB in a chest X-ray. DL-based semantic segmentation mimics radiologists by performing categorization at the pixel level, where each pixel is assigned to a certain case [13]. Using pixel-level labeled data to train the models is referred to as supervised segmentation. Unsupervised or semi-supervised segmentation, in contrast, is a type of segmentation where there are no labeled data [14,15,16]. Weakly supervised learning is in between the supervised and unsupervised learning techniques. It needs a modest amount of labeled data rather than a large quantity. Most of the time, image-level labeling is adequate for weakly supervised learning.

Supervised deep learning models are drawing a lot of interest since they enhance performances and deliver encouraging outcomes in medical fields [17,18,19,20,21,22,23,24,25]. However, the cost of collecting data and annotating them in a clinical setting is substantial. This is due to the lack of doctors with the required expertise, the expensive annotation software, and the lengthy (days to years) data collection process. The survey conducted in [26,27] discussed that the accurate segmentation of medical images using convolutional neural networks depends greatly on labeled data. Nevertheless, given the significant expenses associated with data collecting and labeling, it is very uncommon to create numerous datasets with high-quality labels, particularly in the field of medical image analysis. As a result, much research using incomplete or flawed datasets is needed.

The primary purpose of this literature review is to identify the techniques and procedures used in the weak localization and segmentation of pulmonary tuberculosis manifestations in a chest X-ray image. We identified and highlighted the advantages of these procedures, as well as the problems in detecting pulmonary tuberculosis using these methodologies. The methods for addressing the challenges of the weak localization approach are also discussed in detail. In addition, we discovered and listed publicly available datasets that have at least image-level annotation.

The rest of this paper is structured as follows. In Section 2, we provide the related works and discuss their contributions and shortcomings. In Section 3 of the paper, we discuss our methodology. In Section 4, we provide a list of the available open datasets for tuberculosis localization on chest X-rays. The weakly supervised methods for localizing tuberculosis on a chest X-ray are analyzed in Section 5. Section 6 provides results of our findings, a summary of current weakly supervised technique limitations, and future research directions.

## 2. Related Work

A few publications have investigated and reviewed a few strategies from the standpoint of tuberculosis detection from chest X-ray images [28,29,30,31,32], despite the quantity and scope of systematic evaluations on the weak localization of tuberculosis manifestation in chest X-rays being relatively limited. Hence, this section discusses the most recent surveys and reviews of machine learning’s use in tuberculosis detection from a chest X-ray.

E. Çallı et al. [28] examined the DL-based articles on chest X-rays that were released before March 2021. All of the publications that used classification, regression, segmentation, localization, image generation, and domain adaptation were considered in the review. The authors discussed a paper [33] published in 2019 regarding weakly supervised learning in tuberculosis detection from a chest X-ray.

An overview of deep learning’s advancements for tuberculosis detection from chest X-rays is given in [29]. The writers reviewed publications published on PubMed and Web of Science between 2016 and 2021. Their review’s objectives were identifying existing datasets, methodical contributions, and obstacles and highlighting prospective techniques. They also reviewed studies to find those that offer more than just binary classification for TB, like localization of the region of interest.

The review written by M. Singh et al. [30] pointed out the drawbacks of traditional TB diagnostic methods and gave a study of several machine learning algorithms and their uses in TB diagnosis. The authors identified the lack of good data and their scarcity as the main obstacle to AI application in TB diagnosis. Additionally, they described how DL integrated with neuro-fuzzy, genetic algorithms and how artificial immune systems work better since they consider biological information. The author reviewed the drawbacks of the traditional TB diagnostic methods, identified the challenges, and discussed how other techniques like neuro-fuzzy, genetic algorithms, etc., can improve DL performance. However, they did not highlight the use of weakly supervised segmentation in pulmonary TB screening and only two databases were considered.

DL has the drawback of often only using one modal for the model, which is an image. In contrast, clinical data from medical practice, including demographics, patient evaluations, and lab test results, are also used for TB analysis in addition to pictures. Different DL approaches using single-modal or multimodal algorithms that merged clinical data with images are described in the review in [31]. The authors conducted a thorough search for original studies utilizing DL to identify pulmonary tuberculosis on Springer, PubMed, ResearchGate, and Google Scholar. But the authors did not adequately investigate the use of weak localization in pulmonary TB screening.

## 3. Methodology

This section presents the methodology employed for reviewing weakly supervised segmentation and localization techniques for the diagnosis of pulmonary TB. The systematic literature review (SLR) [34,35] methods are used as a guideline to perform the review. SLR describes a methodical procedure for reviewing relevant material on a subject. The process includes the information obtained in the literature and how and where you searched for it, what search tactics you employed, and the tools you used to find it. SLR uses precise procedures to determine what can be reliably claimed based on these studies to locate as much relevant research on a certain research subject as possible. The pseudo-code Algorithm 1 depicts the steps followed to complete the systematic review as per the PRISMA guideline.
**Algorithm 1:** The steps followed to complete the review as per the PRISMA guideline.1:Formulating review questions: The first step in conducting a systematic review is to identify and formulate the review questions.2:Generating search queries: After formulating the review questions, relevant keywords are selected to generate search queries that will yield suitable results.3:Selecting electronic databases: Once the search queries have been created, appropriate electronic databases are identified and selected to perform the search.4:Defining inclusion and exclusion criteria: Inclusion and exclusion criteria are set and defined to ensure that the retrieved articles are relevant to the review questions.5:Filtering irrelevant articles: Titles and abstracts of the retrieved articles are screened to eliminate duplicates and irrelevant articles from the pool of papers.6:Analyzing relevant articles: The remaining articles are analyzed using all the filtering techniques in accordance with the review questions.7:Reporting and evaluating the systematic review: Finally, the systematic review is reported and evaluated based on the results obtained from the analyzed articles.

### 3.1. Research Questions

This review paper aims to identify weakly supervised learning methods for the localization and segmentation of pulmonary tuberculosis in a chest X-ray. Additionally, it aims to identify the most recent cutting-edge techniques that have been applied in the past five years. Consequently, the main research topic that we hope to address is the following:

PRQ: “What are the state-of-the-art weakly supervised learning techniques utilized for the problem of localizing and segmenting pulmonary tuberculosis in a chest X-ray in the past 6 years?”

Secondary questions are also produced to further assist in focusing the intended response on the primary review question. These secondary questions are provided below:RQ1: Are there any freely available labeled chest radiography datasets? What are their characteristics?RQ2: What are the challenges in accurately localizing pulmonary tuberculosis in a chest X-ray?RQ3: What are the weakly supervised learning techniques in localizing pulmonary tuberculosis in a chest X-ray?RQ4: What are the limitations and challenges in the weakly supervised localization of pulmonary TB in a chest X-ray?

### 3.2. Search Strategies

The primary study’s search approach was centered on six electronic search databases: Google Scholar, PubMed, Science Direct, IEEE Xplore, MDPI, and Springer Link. The search focused on the studies published in the past six years, between 2017 and 2023. All of the database searches were performed from 25 November 2022 to 1 February 2023. But, to obtain a relevant result, we first created a concise search string. Because each electronic search database specifies its syntax, a search string is composed of several keywords and synonyms that are connected using the AND and OR boolean operators. The complete procedure for looking for pertinent works is shown in Algorithm 2.

(Localization OR Segmentation OR Explainable OR Interpretable) OR (Detection OR Identification OR Diagnosis OR Screening) AND (“Chest X-ray” OR “Chest Radiography”) AND (Pulmonary OR Tuberculosis OR TB) AND (“Deep Learning” OR “Weakly Supervised” OR CNN OR “Neural Network”)
**Algorithm 2:** The pseudo-code used to create search strings.1:Databases ← [Google Scholar, Springer Link, PubMed, MDPI, Science Direct, IEEE Xplore]2:**Initialize Keywords**3:Disease_Keywords ← [Pulmonary, Tuberculosis, TB]4:Aim_Keywords ← [Localization, Segmentation, Explainable, Interpretable, Identification, Detection, Diagnosis, Screening]5:Method_Keywords ← [Deep Learning, Weakly Supervised, CNN, Neural Network]6:Data_Keywords ← [Chest X-ray, Chest Radiography]7:Search_String ← “”8:**for** 
disease ∈ Disease_Keywords 
**do**9:    **for** aim ∈ Aim_Keywords **do**10:        **for** data ∈ Data_Keywords **do**11:           Search_String = disease **AND** aim **AND** data12:           **for** database ∈ Databases **do**13:               papers ← databases.search(Search_String)14:           **end for**15:        **end for**16:    **end for**17:**end for**

As shown in the flow diagram in Figure 3, the final result was reached after applying the search string to each individual database and reducing the search results based on the search criteria in Table 1. After performing an advanced search on each of the search engines and applying the inclusion and exclusion criteria, 35 papers were identified for the review purpose. Figure 4 shows that 11 publications were obtained from Springer, 7 from Google Scholar, 6 from IEEE, 5 from PubMed, 4 from Science Direct, and 2 from MDPI.

## 4. Publicly Available Datasets for Tuberculosis Localization in Chest X-ray

As we covered in Section 1 above, chest X-ray images are frequently utilized to identify pulmonary tuberculosis. Several computer-aided PTB detection methods involving chest X-rays have also been developed in recent years. Therefore, it is essential to highlight the characteristics and importance of the most relevant CXR datasets. In this section, we will go over a number of chest X-ray image datasets that are freely accessible to researchers and provide their summary in Table 2:**NIAID** [36] contains 1299 instances in total from five different countries (Azerbaijan, Belarus, Moldova, Georgia, and Romania), 976 (75.1%) of which are multi- or extensively drug-resistant and 38.2% of which contain X-ray images.**TBX11K** [37] has 11,200 CXR images with TB-related bounding box annotations. The pixel size of the images in the dataset is 512 × 512 with five categories: healthy, sick but non-TB, active TB, latent TB, and uncertain TB. Additionally, images from [38,39] are included in the dataset.**CheXpert** [40] is a large public dataset consisting of 224,316 chest radiographs from 65,240 patients that have been classified as positive, negative, or ambiguous based on the presence of 14 observations (atelectasis, cardiomegaly, consolidation, edema, pleural effusion, pneumonia, pneumothorax, enlarged cardiom, lung lesion, lung opacity, pleural other, fracture, support devices, and no findings).**VinDr-CXR** [41] is a dataset containing 100,000 images collected from two major hospitals in Vietnam. Out of this raw data, the publicly released dataset has 18,000 images carefully annotated by 17 expert radiologists, with 22 local labels for rectangles enclosing anomalies and 6 global labels for suspected diseases. The dataset is further divided into 15,000 training sets and 3000 test sets. The test sets’ scans were labeled by the collective opinion of five radiologists, as opposed to the training set’s scans, which were each independently labeled by three radiologists.**MIMIC-CXR** [42] comprises 227,835 images for 65,379 patients who visited the emergency room at Beth Israel Deaconess Medical Center between 2011 and 2016. It is the biggest dataset, containing a semi-structured free-text radiology report that contains the radiological findings of the images and was created contemporaneously during ordinary clinical care.**Shenzhen** [38] contains 662 CXR images, comprising 326 images for normal cases and 336 images depicting TB. These CXR images, which include pediatric CXR, were all taken within a month.**Montgomery** [38] has 138 frontal CXR images, comprising 58 images showing TB and 80 for normal cases. It was gathered in association with the Montgomery County and the US Department of Health and Human Services.**Indiana** [43] is a publicly accessible dataset gathered from various hospitals and provided to the University of Indiana School of Medicine. It contains 7470 CXR images (frontal and lateral) and 3955 related reports. The CXR images in this collection depict a number of disorders, including pleural effusion, cardiac hypertrophy, opacity, and pulmonary edema.**ChestX-ray8** [44] contains 108,948 posterior CXR images in total, of which 24,636 have one or more anomalies. The remaining 84,312 images are from normal patients. The dataset was gathered between 1992 and 2015. Every image in the collection can have many labels, and the labels are for eight different diseases, including pneumothorax, cardiomegaly, effusion, atelectasis, masses, pneumonia, infiltration, and nodules. Natural language processing (NLP) algorithms are used to text-mine the labels from the related radiological reports.**ChestX-ray14** [44] is an updated ChestX-ray8 dataset that includes six additional frequent chest abnormalities (hernia, fibrosis, pleural thickening, consolidation, emphysema, and edema). ChestX-ray14 has 112,120 frontal view CXR images from 30,805 different patients, of which 51,708 have one or more abnormalities and the remaining 60,412 do not. Using NLP methods, ChestX-ray14 was also labeled.**Pediatric-CXR** [45] is made up of 5856 chest X-ray scans of pediatric patients from the Guangzhou Women and Children’s Medical Center in China, 1583 of which are normal cases and 4273 of which have pneumonia.**Padchest (Pathology detection in chest radiographs )** [46] was gathered from 2009 to 2017. There are 168,861 CXR images in it, which were gathered from 67,000 patients at the San Juan Hospital in Spain.**Japanese Society of Radiological Technology (JSRT)** [47] was gathered in 1998 by the Japanese Society of Radiological Technology in coordination with the Japanese Radiological Society from 13 Japanese institutions and 1 American institution. There are 247 posteroanterior CXR images in all, comprising 93 non-nodule CXR images, 100 CXRs with malignant nodules, and 54 with benign nodules. Data from JSRT include metadata, such as the nodule location, gender, age, and nodule diagnosis (malignant/benign). The CXR image size is 2048 × 2048 pixels with a spatial resolution of 0.175 mm/pixel and 12-bit gray levels.**RSNA-Pneumonia-CXR** [48] was gathered by the Radiological Society of North America (RSNA) and the Society of Thoracic Radiology (STR). There are 30,000 CXR images in the dataset in total, of which 15,000 have pneumonia or other related disorders like infiltration and consolidation identified.**Belarus dataset** [49] is a CXR dataset compiled for a study on drug resistance started by the National Institute of Allergy and Infectious Diseases, Ministry of Health, Republic of Belarus. It comprises 306 CXR images of 169 patients.

**Table 2 sensors-23-06781-t002:** Publicly available CXR datasets.

Dataset	Quantity and Size in Pixels	Cases/Findings	Pixel-Level/ Bounding Box Annotation
NIAID [36]	496	TB	N/A
TBX11K [37]	11,200 images with 512 × 512 pixels	Healthy, sick but non-TB, active TB, latent TB, and uncertain	N/A
CheXPert [40]	224,316	Atelectasis, cardiomegaly, consolidation, edema, pleural effusion, pneumonia, pneumothorax, enlarged cardiom, lung lesion, lung opacity, pleural other, fracture, support devices, and no findings	N/A
VinDr-CXR [41]	18,000	Aortic enlargement, atelectasis, cardiomegaly, calcification, clavicle fracture, consolidation, edema, emphysema, enlarged PA, interstitial lung disease (ILD), infiltration, lung cavity, lung cyst, lung opacity, mediastinal shift, nodule/mass, pulmonary fibrosis, pneumothorax, pleural thickening, pleural effusion, rib fracture, other lesions, lung tumor, pneumonia, tuberculosis, other diseases, chronic obstructive pulmonary disease (copd), and no finding	Available
MIMIC-CXR [42]	227,835 images with 2544 × 305 pixels	14 findings	N/A
Montgomery [38]	138 images with 4020 × 4892 pixels	TB and normal	N/A
Shenzhen [38]	662 images with 3000 × 3000 pixels	TB and normal	N/A
Indiana [43]	7470 images with 512 × 512 pixels	10 findings including opacity, cardiomegaly, pleural effusion, and pulmonary edema	N/A
ChestX-ray8 [44]	108,948 images with 1024 × 1024 pixels	Pneumothorax, cardiomegaly, effusion, atelectasis, mass, pneumonia, infiltration, and nodule	Available
ChestX-ray14 [44]	112,120 images with 1024 × 1024 pixels	Pneumothorax, cardiomegaly, effusion, atelectasis, mass, pneumonia, infiltration, nodule, hernia, fibrosis, pleural thickening, consolidation, emphysema, and edema	N/A
Pediatric-CXR [45]	5856	Normal, bacterial pneumonia, viral pneumonia	N/A
Padchest [46]	168,861	193 findings	N/A
JSRT [47]	247 with 2048 × 2048 pixels	Non-nodule, malignant nodules, and benign nodules	Available
RSNA-Pneumonia-CXR [48]	30,000	Pneumonia, infiltration, and consolidation	N/A
Belarus dataset [49]	306 with 2248 × 2248 pixels	TB and normal	N/A

## 5. Weakly Supervised Segmentation and Localization

Fully semantic segmentation can enhance the ability to recognize radiographic characteristics in a CXR and support the diagnosis of pulmonary TB. However, it requires pixel-level annotation. Providing pixel-level annotation for medical images is expensive due to the limited availability of medical imaging data and the need for domain expertise. As a result, we must build a model that uses little to no pixel-by-pixel annotation data. Weakly supervised semantic segmentation helps overcome these issues by providing a technique by which we obtain the pixel-level labels (semantic segmentation) from image-level labels. For instance, the clustering algorithms require no pixel-level labeling and can reach the final cluster without supervision or with partial supervision [50].

A heatmap is a graphical representation of data that depicts the localized region in a weakly supervised localization by representing the individual values included in a matrix as colors. In the context of weakly supervised localization, the heatmap represents the probability of pathology manifested at each pixel location in an image. In the examples we mention in the next section, red indicates the highly significant region, and the other color indicates the less important pixels. In the next section, we will discuss five categories of weak localization techniques: self-training, graphical model based, variants of multiple-instance learning (MIL), weak localization by extracting visualization from a classification task, and seeding-based weakly supervised segmentation. We will also discuss the studies conducted on PTB detection using these techniques.

### 5.1. Self-Training Weakly Supervised Segmentation

The fundamental idea behind self-training is to employ an existing model, known as a pseudo-labeler ($F_{pl}$), to generate predictions on a large dataset that is not labeled, and then use these predictions to produce pseudo-labels for the dataset [51,52,53]. The model is then retrained using the pseudo-labels, with additional regularization [54]. The general loss function of self-training models is given in Equation (Equation 1) as discussed in [54]. If the dataset adheres to the presumption that two examples of the same class can be connected by a series of edges in a graph, self-training on unlabeled data can increase accuracy. Nevertheless, one drawback of the analysis is that it only functions with fine-grained classes, where each class constitutes a connected component of its own in the augmentation graph. The presumptions might not support the situations where a single large class contains weakly related smaller sub-classes and the pseudo-labeler is already underperforming.
(1)$$L=\frac{1}{n}\sum_{m=1}^{n}\sum_{i=1}^{C}L\left(y_{i}^{m},f_{i}^{m}\right)+\alpha \left(e\right)\frac{1}{n^{\prime }}\sum_{m=1}^{n^{\prime }}\sum_{i=1}^{C}L\left(y_{i}^{\prime m},f_{i}^{\prime m}\right)$$
where *n* and $n^{\prime }$ represent the number of samples in labeled data and unlabeled data, respectively; *C* stands for the number of classes. $f_{i}^{m}$ and $f_{i}^{\prime m}$ stand for output of labeled data and unlabeled data, respectively; $y_{i}^{m}$ and $y_{i}^{\prime m}$ correspond to the labeled data label and pseudo-label, respectively; and α(e) represents the balancing coefficient at epoch e.

Self-training weakly supervised segmentation is not used in various applications, such as medical image segmentation, object instance segmentation, and semantic segmentation, where obtaining pixel-level annotations is expensive or impractical. It has other limitations, such as the potential for error accumulation due to noisy pseudo-labels and the need for the careful selection of the stopping criterion to avoid overfitting. Nevertheless, it is an effective approach for training segmentation models when only weak supervision is available.

### 5.2. Graphical-Model-Based Weakly Supervised Segmentation

Images can be represented in a graph by connecting pixels to their neighboring pixels. The vertex set of the graph represents image elements and its edge set is given by an adjacency relation to the image elements. There are two relations used to create a graph from an image: pixel adjacency and region adjacency. A pixel adjacency graph connects similar pixels, whereas a region adjacency graph connects regions with similar properties. The homogeneity of superpixels, which are the outcome of the perceptual grouping of pixels [55,56], can be captured via spatial structure modeling, and graphical models can be used to address problems that arise in weakly supervised segmentation, such as a lack of association. Graphical models provide a framework for representing and modeling the relationships between variables in a probabilistic graphical manner, which is used for capturing spatial dependencies and incorporating prior knowledge into the segmentation process.

One common type of graphical model used in weakly supervised segmentation is the Markov Random Field (MRF) or Conditional Random Field (CRF) [57,58]. These models allow for the modeling of spatial dependencies between adjacent superpixels or pixels, capturing the homogeneity of neighboring regions. MRFs and CRFs can incorporate both local and global contextual information, such as color, texture, and spatial proximity, to guide the segmentation process. Graphical models incorporate prior knowledge or constraints, such as shapes or object sizes, into the segmentation process. This can help improve the accuracy and robustness of the segmentation results, even when there is a lack of fully supervised training data.

In [59,60], a semantic association between sets of superpixels is learned from image-level labels. The learned semantic association guides the semantic labeling of each superpixel. The authors in both papers attempted to capture the spatial organization of superpixels by extracting graphlets [61] and linking nearby superpixels. The graphlets are small connected sub-graphs representing the high-order structural potentials among superpixels. Image-level labels, the global spatial layout, the geometric context, and multichannel appearance attributes are then encoded into graphlets via patch-alignment-based embedding. The graphlet’s size is equal to t if the number of its constituent superpixels is t. The t sized graphlet can be represented using a t x t sized matrix $M_{s}$ with Equation (Equation 2).
(2)$$M_{s}\left(i,j\right)=\left\{\begin{array}{c} \theta (R_{i},R_{j}),\;\mathrm{if}\;R_{i},\;\mathrm{and}\;R_{j}\;\mathrm{are}\;\mathrm{spatially}\;\mathrm{adjacent}\\ 0,\;\mathrm{otherwise} \end{array}\right.$$
where θ($R_{i}$,$R_{j}$) is the angle between the horizontal axis and the vector from the center of superpixel $R_{i}$ to the center of superpixel $R_{j}$.

Both color and texture attributes are employed to capture the appearance of the superpixel that makes up a graphlet. The texture channel of the superpixel in the graphlet is represented by a 128-dimensional histogram of gradient (HOG) [62]. Hence, the texture matrix for the graphlet will be $M_{R}^{T}$, a tx128 matrix. The nine-dimensional color moment [63] is also extracted for each superpixel in the graphlet and produces a tx9 $M_{R}^{C}$ matrix. The texture color channel of a graphlet is given by $M^{T}$ = [$M_{R}^{T}$, $M_{S}$] and $M^{C}$ = [$M_{R}^{C}$, $M_{S}$]. Two identical-sized graphlets in either the texture or color channel can be compared using the Golub–Werman distance given in Equation (Equation 3), where $M_{0}$ and $M_{0}^{\prime }$ are the respective orthonormal bases of two identically sized matrices M and $M^{\prime }$, which are given by their respective matrices as points on the Grassmann manifold.
(3)$$d_{GW}\left(M,M^{\prime }\right)={\left\|M_{0}-M_{0}^{\prime }\right\|}_{2}$$

The authors in [64] proposed a Graph Regularized Embedding Network (GREN), which uses the cross-image and cross-region information to localize radiographic features on chest X-ray images. The GREN tries to model the anatomical structures of the chest X-ray by taking both the relationships between and within images into account. In order to enable the deep embedding layers of the neural network to keep structural information, the graphs are computed using the regularizers hash coding and Hamming distance.

Assuming the chest X-ray dataset is represented by *X* = $\{X_{1}$, $X_{2}$,…, $X_{n}\}$ and segmented lung lobes by $X^{\prime }$ = $\{X_{1}^{l}$, $X_{1}^{r}$, $X_{2}^{l}$, $X_{2}^{r}$, …, $X_{n}^{l},X_{n}^{r}\}$, the feature extractor (CNN) in Figure 5 produces feature maps $F_{inter}$ = $\{X_{1}$, $X_{2}$,…, $X_{n}\}$ and $F_{intra}$ = $\{f_{1}^{l}$, $f_{1}^{r}$, $f_{2}^{l}$, $f_{2}^{r}$,…, $f_{n}^{l},\;f_{n}^{r}\}$, respectively, for the input *X* and $X^{\prime }$. The objective function Q(θ) to be optimized is given by Equation (Equation 4) where *L*(θ) represents the loss function for the localization network and *D*(θ) is for the graph regularization computed from the inter-image and intra-image relationship.
(4)$$Q\left(\theta \right)=L\left(\theta \right)+\lambda_{1}D{\left(\theta \right)}_{intra}+\lambda_{2}D{\left(\theta \right)}_{inter}$$

The loss function for the GREN network is given in Equation (Equation 5). It has two loss functions: $L_{i}^{k}{\left(\theta \right)}_{1}$ (Equation (Equation 6)) for when there are pixel-level annotations for the disease region and $L_{i}^{k}{\left(\theta \right)}_{2}$ (Equation (Equation 7)) for when there are none.
(5)$$L\left(\theta \right)=\sum_{i}\sum_{j}\beta \lambda_{i}^{k}L_{i}^{k}{\left(\theta \right)}_{1}+\left(1-\lambda_{i}^{k}\right)L_{i}^{k}{\left(\theta \right)}_{2}$$
(6)$$L_{i}^{k}{\left(\theta \right)}_{1}=\sum_{j}-y_{ij}^{k}log\left(p_{ij}^{k}\right)-\sum_{j}\left(1-y_{ij}^{k}\right)log\left(1-p_{ij}^{k}\right)$$
(7)$$L_{i}^{k}{\left(\theta \right)}_{2}=-y_{i}^{k}log\left(1-\prod_{j}\left(1-p_{ij}^{k}\right)\right)-\left(1-y_{i}^{k}\right)log\left(\prod_{j}\left(1-p_{ij}^{k}\right)\right)$$

Equation (Equation 9) is used to analyze the relationship between images of the left ($X_{i}^{l}$) and right ($X_{i}^{r}$) lobes of a lung, whereas Equation (Equation 8) analyzes the relationship among images of the chest X-ray ($X_{u}$, $X_{v}$). The graph $G_{intra}$ = (V, E), where the vertex V stands for the lung lobes and the edge $e_{i}^{lr}$∈ E stands for the similarity between the graph vertex V, represents the structural relationship between the lung lobes of a chest X-ray. To compute the structural relationship between a randomly selected pair of images, $X_{u}$ and $X_{v}$, randomly from the input chest X-ray images X, we use a graph $G_{inter}$ = (V, E), where the graph nodes V denote the images in a mini-batch of the network and the edges $e_{uv}$∈ E denote the similarity of graph nodes V. *d*(∗) in both (Equation 8) and (Equation 9) is the distance metric function, which is the Euclidean distance.
(8)$$D{\left(\theta \right)}_{inter}=-\sum_{u,v}e_{uv}d\left(f_{u},f_{v}\right)$$
(9)$$D{\left(\theta \right)}_{intra}=-\sum_{i}e_{i}^{lr}d\left(f_{i}^{l},f_{i}^{r}\right)$$

Overall, graphical models can be powerful tools in weakly supervised segmentation, as they provide a flexible framework for modeling the spatial structure and incorporating prior knowledge, which can lead to more accurate and robust segmentation results. However, like any other method, they have certain limitations [65,66]. The approach often requires initialization, such as the selection of seeds or superpixels, to start the segmentation process. The performance of these methods can be sensitive to the quality of the initialization, which may affect the accuracy and robustness of the segmentation results. Choosing the optimal initialization can be challenging and may require trial and error or expert knowledge. It can also be computationally expensive, especially when dealing with large datasets, due to the iterative nature of graph-based optimization algorithms. This can limit their scalability to handle large-scale datasets or real-time applications, as they may require significant computational resources and time. Hence, it is important to consider these limitations when using graph-based weakly supervised segmentation methods and choose appropriate techniques or strategies to mitigate them depending on the specific application requirements and constraints.

### 5.3. Variants of Multiple-Instance Learning (MIL) for Weakly Supervised Segmentation

In image classification tasks, unless the region of interest is specified, the entire image is represented by a single-class label. For example, a CXR image may be labeled as “consolidation” even though only a small portion of the image contains consolidation. Multiple-instance learning (MIL) is a machine learning approach that can be used to address this challenge. MIL models learn to predict a class label for a bag of pixels, where each pixel is an instance. The model also learns to identify which pixels in the bag contributed to the prediction of the class label [67]. This information can be used to produce segmentation masks, which can help to identify the specific areas of an image that belong to a particular class.

Each image is a bag $B_{k}$ of pixels ${\left\{x_{k}^{j}\right\}}^{n}$ with a label $y_{k}$, where *n* is the number of pixels in a bag, j∈{0,1,…,n}, and *k* indicates the *k*th bag in the image. The bag may belong to multiple categories at the same time. Hence, the bag can have multiple labels, $y_{k}\subseteq \left\{1,2,\ldots ,C\right\}$. The bag probability in the MIL context must be permutation-invariant, which means that the order of the pixels in the bag does not matter. The model is only interested in the presence or absence of certain patterns in the bag, not the order in which the patterns appear [68].

A method developed by Li et al. [69] involves partitioning a picture into a set of m patches, after which the *i*th image will become $x_{i}$ = { $x_{i1}$, $x_{i2}$,…, $x_{im}$ }, where *m* = PxP as we will have grids with P sizes. Each patch in the input image receives a probability score according to the entire model. To identify the active patches from the non-activated ones, the score threshold $T_{s}$ is constructed. The *j*th patch in the *i*th image belongs to the localization for class c if the probability scores $p_{ij}^{k}$ are greater than $T_{s}$.

Although it can help provide semantic segmentation with image-level labeling (bag label), it fails to model the interactions between superpixels, which is crucial for smoothing superpixel labels. The other weakness of this weakly supervised segmentation method is the low descriptive pairwise potentials, resulting in many ambiguous segment boundaries. The authors in [70] utilized the fundamental architecture of the Semantic Texton Forest (STF) [71] and extended it for the multiple-instance learning method. The authors in [72] tried to use an attention-based mechanism to provide insight into the contribution of each instance to the bag label.

### 5.4. Weak Localization by Extracting Visualization from Classification Task

The approaches that are primarily used for visualization but also utilized for localization will be covered in this section. Occlusion sensitivity [73], saliency maps [74], class activation maps (CAMs) [75], gradient-weighted class activation maps (Grad-CAMs) [76], Grad-CAM++ [77], Score-CAM [78], class-selective relevance maps (CRMs) [79], and attention networks [80] are the most commonly used visualization techniques for the localization of pulmonary TB in chest X-rays (CXRs), as summarized in Table 3. In addition, these techniques have been used in interpreting the reasoning behind the output of complex black-box models designed for medical fields [81,82,83].

#### 5.4.1. Occlusion Sensitivity

In occlusion sensitivity [73], by methodically covering various areas of the input image with a gray square box and observing the classifier’s output, an attempt is made to determine the contribution of an image portion to the classifier’s output. The occluded region has a significant impact on the prediction if the classifier’s output decreases as a result of covering the region by the box. However, occlusion can be costly if the input image size is high and the box size is small. In the study conducted by M.T. Islam et al. [84], the localization of pulmonary TB manifestation in CXRs is acquired by taking advantage of a classifier model’s occlusion sensitivity.

#### 5.4.2. Saliency Map

As opposed to the occlusion sensitivity method, which disturbs squares of the image, the saliency map [74] approach perturbs each and every pixel. In the research work published by F. Pas et al. [85], saliency map localization techniques were used to visualize and localize pulmonary TB manifestations in chest X-ray images. In the saliency map, the feature visualization is performed by taking the class score gradient with respect to the input image (Equation (Equation 10)) and back-propagating the gradient through the model back to the first layer of the model. It will produce an activation class that depicts the degree to which a pixel contributed to the class score. Pixel attribution approaches can be exceedingly unstable and unreliable since even very small changes to the image could lead to entirely different pixels being highlighted as explanations even though they still produce the same prediction.
(10)$$Grad\left(I_{0}\right)=\frac{\partial S_{c}}{\partial I}{|}_{I=I_{0}}$$

Given an image $I_{0}$, a class *c*, and a classification score $S_{c}$, which is the output of the classifier model, the $Grad(I_{0})$ will calculate the attribution of the pixels of $I_{0}$ on the class score $S_{c}$.

#### 5.4.3. Class Activation Map (CAM)

The CAM [75] does not employ a gradient like a saliency map. The weights of the contribution of feature maps to the class score are calculated using global average pooling (GAP) [86]. Global average pooling outputs the spatial average of each feature map unit in the final convolutional layer (Equation (Equation 11)). The final output is produced by adding the weighted sum of the GAP values (Equation (Equation 12)). Similarly, class activation maps are constructed from a weighted sum of the feature maps (Equation (Equation 13)). Multiple previously reported works have incorporated CAM-based localization. In [87], after the abnormality of CXRs was checked, the author performed further classification to identify the manifestations of pulmonary TB in the abnormal CXR images. Then, the author used a CAM to localize the suspicious disease area on the images. The biggest drawback of the CAM is that it is architecture-sensitive, necessitating the use of a global pooling layer after the convolutional layer of interest.
(11)$$\mathrm{GAP}=\sum_{x,y}f_{k}\left(x,y\right)$$
(12)$$S_{c}=\sum_{k}w_{k}^{c}\sum_{x,y}f_{k}\left(x,y\right)$$
(13)$$M_{c}\left(x,y\right)=\sum_{k}w_{k}^{c}f_{k}\left(x,y\right)$$

$f_{k}\left(x,y\right)$ represents the activation of the *k* feature map in the last convolutional layer at spatial location (*x*,*y*); $\sum_{x,y}f_{k}\left(x,y\right)$ represents GAP; $w_{k}^{c}$ represents the weight corresponding to class *c* for unit *k*; and $M_{c}\left(x,y\right)$ represents the class activation map for class *c*.

#### 5.4.4. Grad-CAM (++)

The Grad-CAM [76] provides a rough localization map by calculating the gradient of the class score with respect to the final convolutional layer (Equation (Equation 14)). Assuming that there are k feature maps in the last convolutional layer, which are represented with $F_{1}$, $F_{2}$,…, $F_{k}$, the Grad-CAM (Equation (Equation 17)) decides the attribution of each of the k feature maps to the class score $S_{c}$. Hence, the feature map is weighted with the gradient, and the average over the feature maps is calculated. Finally, the rough heatmap is overlaid over the input image. The Grad-CAM++ [77] is an upgraded version of the Grad-CAM with the goal of improving object localization and locating instances of multiple items in a single image. The only difference between the Grad-CAM++ and the Grad-CAM is the calculation of weights $w_{k}^{c}$, which is given with Equation (Equation 14) for the Grad-CAM and Equation (Equation 15) for the Grad-CAM++. The Grad-CAM++ is equivalent to the Grad-CAM when $\alpha_{ij}^{kc}$ = $\frac{1}{z}$.
(14)$$w_{k}^{c}=\frac{1}{Z}\sum_{i}\sum_{j}ReLU\left(\frac{\partial S_{c}}{\partial F_{ij}^{k}}\right)$$
(15)$$w_{k}^{c}=\sum_{i}\sum_{j}\alpha_{ij}^{kc}ReLU\left(\frac{\partial S_{c}}{\partial F_{ij}^{k}}\right)$$
(16)$$\alpha_{ij}^{kc}=\frac{\frac{\partial^{2}Y^{c}}{{\left(\partial F_{ij}^{k}\right)}^{2}}}{2\frac{\partial^{2}Y^{c}}{{\left(\partial F_{ij}^{k}\right)}^{2}}+\sum_{a}\sum_{b}F_{ab}^{k}\left(\frac{\partial^{3}Y^{c}}{{\left(\partial F_{ij}^{k}\right)}^{3}}\right)}$$
(17)$$M_{Grad-CAM}^{c}=\sum_{k}w_{k}^{c}F^{k}$$

The Grad-CAM was suggested in [85,88] to enable visual localization in determining the area afflicted by tuberculosis from chest X-ray pictures. The authors of [85] say that they found saliency maps to be more successful than Grad-CAMs since tuberculosis manifests with small features. The Grad-CAM++ is used in [89] for better and more accurate weakly supervised segmentation of tuberculosis from a chest X-ray. The Grad-CAM++ is used to create class activation maps; after that, the heatmaps are thresholded and then refined with a dense conditional random field (CRF) [58] to improve the boundaries between various object classes. The author of the paper went beyond providing visual localization and created segmentation masks utilizing post-processing techniques like noise filtering, thresholding, and CRF, and an example of their finding is given in Figure 6.

The two fundamental problems with gradient-CAM and grad-CAM++ localization are gradient saturation and false confidence. The gradient of the output in relation to feature map activation may appear noisy due to the gradient calculation cost and potential vanishing issues. False confidence is the idea that more input regions that are significant to making decisions are not always produced by activation maps with greater weights.

#### 5.4.5. Score-CAM

The Score-CAM [78] eliminates the need for gradient computations by using weights from forward passing scores on the target class. In [49], T. Rahman et al., used a Score-CAM and the t-Distributed Stochastic Neighbor Embedding (t-SNE) [90] visualization technique to identify the region that contributed the most to the classifications, as depicted in Figure 7. The visual explanation that is based on perturbations is what drives the Score-CAM. However, the author advised masking the highlighted input region to perturb activation maps rather than directly perturbing the input region. The model’s reaction to perturbed input is then used to calculate the weights.

Given a general function Y = *f*(*X*) that takes an input vector *X* = ${\left[x_{0},\;x_{1},\ldots ,\;x_{n}\right]}^{T}$ and produces scalar output Y, for a known baseline input $X_{b}$, the contribution $c_{i}$ of $x_{i}$ towards Y is the change in the output by replacing the *i*th entry in $X_{b}$ with $x_{i}$ (Equation (Equation 18)).
(18)$$c_{i}=f\left(X_{b}⊙H_{i}\right)-f\left(X_{b}\right)$$
where $H_{i}$ is a vector with the same shape as $X_{b}$ but for each entry $h_{j}$ in $H_{i}$, $h_{j}$ = *X* [*i* = *j*], and ⊙ denotes an element wise product.

Hence, for a CNN model Y = *f*(*X*) takes an input *X* and outputs a scalar Y, by picking the convolutional layer *l* in f and the corresponding activation as A, for a known baseline input $X_{b}$, and the contribution $A_{l}^{k}$ towards Y is defined in Equation (Equation 19).
(19)$$C\left(A_{l}^{k}\right)=f\left(X⊙H_{l}^{k}-f\left(X_{b}\right)\right)$$
where $H_{l}^{k}$ can be calculated with Equation (Equation 20), which up-samples $A_{l}^{k}$ into the input size with Up(·) and normalizes each element in the input matrix into [0, 1] using s(·).
(20)$$H_{l}^{k}=s\left(Up\left(A_{l}^{k}\right)\right)$$

#### 5.4.6. Class-Selective Relevance Map (CRM)

The CRM was first introduced in [79] for providing a visual interpretation of CNN predictions in classifying medical images. Later on, it was used for pulmonary TB localization [91,92]. The CRM locates the most discriminating ROI in the input image by determining the significance of the activation of the feature maps at the last convolutional layer of a CNN model at a spatial point (*x*,*y*). The prediction scores $S_{c}$ (Equation (Equation 21)) are first calculated for each output node, and then the prediction scores $S_{c}$(*l*,*m*) (Equation (Equation 22)) are calculated for each output node once the spatial element (*l*,*m*) in the feature maps from the last convolutional layer has been removed.
(21)$$S_{c}=\sum_{x,y}\sum_{k}w_{k}^{c}f_{k}\left(x,y\right)$$
(22)$$S_{c}\left(l,m\right)=\sum_{x,y\neq l,m}\sum_{k}w_{k}^{c}f_{k}\left(x,y\right)$$

The CRM (Equation (Equation 23)) is then described as a linear sum of the incremental MSE between $S_{c}$ and $S_{c}\left(l,m\right)$ computed from each node in the model’s output layer. An important component in the feature maps from the final convolution layer would maximize the difference between these prediction scores by both making a positive contribution to raising the prediction score at the output node representing the desired class and a negative contribution to lowering it at the other output nodes.
(23)$$R\left(l,m\right)=\sum_{c=1}^{N}{\left(S_{c}-S_{c}\left(l,m\right)\right)}^{2}$$

#### 5.4.7. GSInquire

GSInquire [93] uses an inquisitor to probe a set of vertices and edges from a network and observes the responses in order to generate an interpretation of the network’s decision-making process. It was discovered to offer explanations that more accurately reflect the decision-making behavior of deep neural networks and identify critical factors that are quantitatively critical to the decision-making process instead of relative heatmaps [94]. In [95], A. Wong et al. utilized GSInquire to provide a visual explanation and localization of the region affected by pulmonary TB. The radiologist validation process involved comparing these highlighted regions with how the radiologist interpreted the cases, and it was found that in they were similar in all cases.

#### 5.4.8. Attention Mechanism

In CNNs, only local information from the input is used to calculate the output, which may introduce bias because global information is not considered. Additionally, there are potentially simplistic solutions to the issue, such as utilizing more convolution layers or applying multiple-receptive field [23,25,96,97] techniques in deeper networks. Although the outcomes are not noticeably improved, the computational overhead increases. A spatial CNN with an attention mechanism that focuses on the relationships between the critical features on the full input patch and the physically close pixels has been proposed in [98,99,100,101,102]. For the localization of PTB pathology in the chest X-ray, several attention map mechanisms have been put forth in [95,103,104,105,106,107,108].

S. Sedai et al. [103] utilized DenseNet blocks as a base network and developed a classification and attention CNN that combines the intermediate feature maps using learned layer relevance weights, as depicted in Figure 8. The intermediate feature maps are combined by CNN (C-CNN) utilizing learned layer relevance weights. The multiscale attention map is computed using the attention CNN (A-CNN) employing feature maps and learned layer relevance weights.

A contrast-induced attention network (CIA-Net) is presented in the paper [105] to take advantage of the highly structured nature of chest X-ray images and pinpoint TB consistent regions using contrastive learning on aligned positive and negative samples. Additionally, the author includes an alignment module that can be trained to correct all of the input images, which minimizes scale, angle, and displacement changes in X-ray images acquired with subpar scan settings. According to the author, the attention module is forced to concentrate on anomalies as a result. The CIA-Net consists of two branches, as illustrated in Figure 9. The first branch extracts a $F_{i}^{+}$ feature map for a positive image $I_{i}^{+}$ while the second takes both the positive $I_{i}^{+}$ and the negative $I_{i}^{-}$ images as a pair of input and extract attention maps $M_{i}^{+}$ and $M_{i}^{-}$, respectively. The absolute difference ΔM = $|M_{i}^{+}-M_{i}^{-}|$ is calculated and it is used as a spatial attention map. The weighted feature map (WFM) obtained from $F_{i}^{+}$ and ΔM is given in Equation (Equation 24), where $\Delta m_{k}$ stands for the *k*th weight in ΔM and $f_{k}$ stands for the *k*th grid in $F_{i}^{+}$.
(24)$$WFM_{i}^{+}=\sum_{k}^{wxh}\Delta m_{k}f_{k}$$

There are two different kinds of loss functions used in the study. When there is bounding box annotation, one type of loss function is applied; when there is no annotation at all, the other type is applied. If the grid in the feature map overlaps the projected ground truth box in an image with bounding box annotations, the grid is labeled with 1; otherwise, it is labeled with 0. As a result, a binary cross-entropy loss given in Equation (Equation 25) is used for each grid, where *k*, *i*, and *j* are the indexes of classes, datasets, and grids, respectively. $y_{ij}^{k}$ denotes the target label of the grid and $p_{ij}^{k}$ denotes the predicted probability of the grid. For images with only image-level annotations, the MIL loss suggested in [69] is applied, as given in Equation (Equation 26), where $y_{i}^{k}$ denotes the target label of the image.
(25)$$L_{i}^{k}\left(B\right)=\sum_{j}-y_{ij}^{k}log\left(p_{ij}^{k}\right)-\sum_{j}\left(1-y_{ij}^{k}\right)log\left(1-p_{ij}^{k}\right)$$
(26)$$L_{i}^{k}\left(I\right)=-y_{i}^{k}log\left(1-\prod_{j}\left(1-p_{ij}^{k}\right)\right)-\left(1-y_{i}^{k}\right)log\left(\prod_{j}\left(1-p_{ij}^{k}\right)\right)$$

### 5.5. Seeding-Based Weakly Supervised Segmentation

Classification neural networks can produce weak object localization (seed cues). However, they cannot predict the precise spatial extent of objects. Most of the works regarding the generation of seed cues rely on prediction gradients with respect to the input data from a trained classifier [76]. Some studies propose to train GAP-based classifiers [75] without dealing with gradient calculation. A process that generates cues of object localization in [109] is called weak localization. Even though localization cues cannot serve as complete and accurate segmentation masks, they can significantly benefit the weakly supervised segmentation network. The work in [109] aimed at training a convolutional neural network f(X:θ), parameterized by θ, that models the conditional probability of observing any label *c*∈*C* at any location *u*∈ {1, 2,…, *n*}, i.e., $f_{u,c}\left(X;\theta \right)=p\left(y_{u}=c|X\right)$. A method for determining parameters, θ, for the segmentation neural network depends on minimizing a three-term loss function. The first term ($L_{seed}$) provides the localization hint to the model; the second term ($L_{expand}$) penalizes the model for wrong object prediction and too small segmentation masks. Finally, the third term ($L_{constrain}$) encourages segmentation that respects the spatial and color structure of the images.

In [110], the author proposed an easy-to-hard principle that masters knowledge with increasing difficulty. A model is first trained on easy samples and then on more difficult samples. This principle is called curriculum learning [111]. It begins with a CAM creating a heatmap for the TB consistent region. The heatmap is then refined using attention-guided iterative refinement. The addition of heatmap regression forces the learning of superior convolutional features under the guidance of attention and produces more accurate heatmaps. This is because the heatmap regression loss (Equation (Equation 27)) penalizes the model for predicting incorrect heatmaps.
(27)$$Loss_{reg}\left(I\right)=\sum_{d=1}^{D}\sum_{x,y}smooth_{L_{1}}\left({\hat{H}}^{c}\left(x,y\right)-H^{c}\left(x,y\right)\right),\forall I\in S$$

First, the seed attention map from the previous iteration ($H^{c}$) and the heatmap produced by the current network (${\hat{H}}^{c}$) are subtracted from each other, where c is the positive disease category, c∈{1,…,C}. Then, as shown in Equation (Equation 28), smooth $L_{1}$ losses are imposed over the feature channel. The total loss value will be calculated by adding all of the losses over the feature channel *D*, where d∈{1,…,D}.
(28)$$smooth_{L_{1}}\left(z\right)=\left\{\begin{array}{c} 0.5z^{2},\;\mathrm{if}\;\left|z\right|\;<\;1\\ |z|-0.5,\;\mathrm{otherwise} \end{array}\right.$$

The final objective function that needs to be optimized is given in Equation (Equation 29). It combines the sigmoid cross-entropy loss for multilabel classification ($Loss_{cls}$) and the heatmap regression loss ($Loss_{reg}$) for localization. If an image (*I*) has a seed for a disease category *c* ∈ *C*, then $b_{c}$ = 1; otherwise, 0. The classification and regression losses are balanced using λ to make both losses make roughly equal contributions.
(29)$$Loss_{final}\left(I\right)=Loss_{cls}\left(I\right)+\lambda \sum_{c=1}^{C}b_{c}Loss_{reg}\left(I\right)$$

The equations above show that there are situations where there are no seeds. This is because the seeds (abbreviated as S) are only harvested from severe and moderate illness severity levels, where the outward manifestations of the illness are more easily distinguishable than in mild cases.

## 6. Discusion

Using the search strategies described in Section 3.2, we extracted several relevant scholarly works focusing on the weakly supervised localization and segmentation of pulmonary tuberculosis in the CXR images. Among these retrieved scholarly articles, thirty-five (35) papers were selected by applying the inclusion and exclusion criteria shown in Table 1. Summaries of these papers in terms of the publication year and techniques employed are provided in Figure 10 and Figure 11, respectively.

Semantic segmentation can assist physicians by localizing diseased areas and determining the region of interest in a chest X-ray image. But training a fully supervised algorithm is a challenging task that consumes expert physicians’ time. Hence, weakly supervised segmentation comes to the rescue where there are only image-level labels but not pixel-level labels. In Section 5, we discussed techniques commonly used to perform weak localization and weakly supervised segmentation. In addition, we summarized these techniques and datasets used for the weak localization of PTB in chest X-ray images on Table 4. Most of the works focus on using heatmaps for the visualization and localization of the PTB disease manifestation regions on chest X-ray images and explaining the output of DL models.

We identified five major categories of disease localization methods in studies, including (1) self-training, (2) graphical models, (3) multiple-instance learning, (4) weak localization by extraction of visualization from classification task, and (5) seeding-based weakly supervised segmentation. The studies in [44,64,69,105,107,110,112] proposed approaches that are clear, robust, and efficient in detecting tuberculosis-consistent regions. These methods utilize bounding boxes to improve their performance and present their results using the intersection over the bounding box (IoBB) metric over the ChestX-ray8 and ChestX-ray14 datasets. The multiple-instance learning in [69] and the hierarchical attention in [110] achieve state-of-the-art performances with 78% and 85% accuracy at T(IoBB) = 0.1, respectively, where T(*) is the threshold. So, the localization is correct when the IoBB exceeds the T(IoBB).

The performance of weakly supervised segmentation techniques can be enhanced by incorporating additional information sources, self-supervised learning techniques in the pipeline, iterative or multistage procedures, regularization techniques, dealing with noise and low-quality data, and fine-tuning the model architecture. Most notably, addressing noise and poor data quality is critical in improving poor localization performance. Pre-processing techniques, such as data cleaning, noise reduction, and data augmentation, can assist in reducing the influence of noisy or low-quality data on segmentation outcomes.

Integrating additional sources of information and domain-specific knowledge can also help to improve the accuracy of segmentation results. For example, implementing image-level labels and bounding box annotations can provide complementary information and lead to more accurate segmentation results. Iterative or multistage approaches that involve refining the initial segmentation result in multiple iterations or stages can also improve the segmentation result. For example, using an initial weakly supervised segmentation as a starting point and then refining it with additional steps, such as post-processing techniques or fine-tuning with a small amount of labeled data, can help improve the accuracy of the final segmentation results. In addition, proposing new model architectures or modifying existing ones can enhance the performance of weakly supervised segmentation. Furthermore, regularization techniques, such as spatial regularization or consistency regularization, can help improve the robustness and accuracy of weakly supervised segmentation.

These strategies encourage smoothness and consistency in anticipated segmentation maps, resulting in more accurate results. However, it is crucial to note that the efficiency of these strategies may vary depending on the dataset, challenge, and weakly supervised segmentation approach used. So, rigorous experimentation and the study of the weak localization results aid in identifying the most effective ways for enhancing weakly supervised segmentation.

**Table 4 sensors-23-06781-t004:** Summary of Reviewed Papers.

Author and Year	Dataset	Method
Wang et al., 2017 [44]	ChestX-ray8	CAM with Log-Sum-Exp (LSE) [113]
Islam et al., 2017 [84]	Indiana, JSRT, and Shenzhen datasets	Occlusion sensitivity [73]
Seda et al., 2018 [103]	ChestX-ray14	Multiscale attention map and layer relevance weights
Li et al., 2018 [69]	ChestX-ray14	Multiple-instance learning (MIL) [114]
Liu et al., 2018 [115]	Shenzhen, Montgomery, and 2443 frontal chest X-rays from Huiying Medical Technology	CAM [75]
Tang et al., 2018 [110]	ChestX-ray14	CAM [75] with attention-guided iterative refinement
Wang et al., 2018 [104]	ChestX-ray14, Indiana	Multilevel attentions and saliency-weighted global average pooling
Zhou et al., 2018 [116]	ChestX-ray14	Adaptive DenseNet
Pas et al., 2019 [85]	Belarus Tuberculosis Portal, Montgomery, Shenzhen and NIH CXR datasets	Saliency maps [74] and grad-CAMs [76]
Liu et al., 2019 [105]	ChestX-ray14	Attention networks
Rahman et al., 2019 [49]	NLM, Belarus, NIAID TB, and RSNA datasets	Score-CAM [78] and t-Distributed Stochastic Neighbor Embedding (t-SNE) [90]
Guo and Passi, 2020 [87]	Shenzhen and the NIH CXR dataset	Class activation map (CAM) [75]
Singh et al., 2020 [106]	The posteroanterior CXRs from Christian Medical College in Vellore, India	Multiscale attention map
Ouyang et al., 2020 [107]	NIH ChestX-ray14 and CheXpert datasets	Hierarchical attention network [80]
Chandra et al., 2020 [117]	Montgomery	Fuzzy C-Means (FCM) and K-Means (KM)
Viniavskyi, Dobko, and Dobosevych, 2020 [89]	SIIM-ACRPneumothorax	Grad-CAM++ [77], conditional random fields (CRF) [58], and inter-pixel relation network (IRNet) [118]
Rajaraman et al., 2021 [91]	TBX11K CXR dataset	Saliency maps and a CRM-based localization algorithm [79]
Mamalakis et al., 2021 [119]	Pediatric CXR and Shenzhen	Heatmaps [119]
Qi et al., 2021 [64]	Chest-Xray14	Graph-Regularized Embedding Network (GREN)
Wong et al., 2022 [95]	NLM, Belarus, NIAID TB, and RSNA datasets	Visual attention condensers [120] and GSInquire [94]
Rajaraman et al., 2022 [92]	The dataset contains 224,316 CXRs collected from 65,240 patients at the Stanford University Hospital in California.	Class-selective relevance maps (CRMs) [79]
Nafisah and Muhammad, 2022 [88]	Montgomery, Shenzhen, and Belarus CXH datasets	Grad-CAM [76] and t-SNE visualization technique [90]
Bhandari et al., 2022 [121]	Shenzhen, Montgomery, and Belarus datasets	LIME [122], SHAP [123], and Grad-CAM [76]
Mehrotra et al., 2022 [124]	Indiana, Shenzhen, and Montgomery	Class activation map (CAM) [75]
Zhou et al., 2022 [125]	Shenzhen and Montgomery	Heatmap
Rajaraman et al., 2022 [92]	CheXpert CXR and PadChest CXR datasets	Class-selective relevance maps (CRMs) [79], and attention maps [126]
Visuña et al., 2022 [127]	COVID-QU-Ex, NIAID, Belarus, RSNA Pneumonia, Shenzhen, and Montgomery	Grad-CAM [76]
Prasitpuriprecha et al., 2022 [128]	Shenzhen and Montgomery	Grad-CAMs [76]
Souza et al., 2022 [112]	Chest-Xray14	CAM refined with PCM [129]
Malik et al., 2022 [130]	RSNA, Chest-Xray14, and other COVID-19 CXRs	Grad-CAMs [76]
Tsai et al., 2022 [131]	Montgomery, Shenzhen, Tbx11k, and Belarus	Grad-CAMs [76]
Li et al., 2022 [132]	CheXpert	GL-MLL
Pan et al., 2022 [108]	In addition to TBX11K, the author also prepared TBX-Att dataset from TBX11K	Multihead cross attention with attribute reasoning
Kazemzadeh et al., 2023 [133]	Indiana, Shenzhen, Montgomery, and other data collected from different countries	Grad-CAMs [76]

## 7. Conclusions

This work identified the most recent weakly supervised semantic segmentation techniques and how these methods have been used for localizing pulmonary tuberculosis from chest X-ray images. To meet this objective, we developed search strategies based on the identified research questions. We used six online search databases to find the works that address the research questions and meet the selection criteria indicated in Table 1. After rigorously searching and applying the selection criteria, we identified 35 research articles published between 2017 and 2023. These investigations allowed us to identify five commonly used methods for weakly supervised localization: self-training, graphical models, MIL variations, visualization, and seeding-based weakly supervised segmentation. Most studies on the localization of pulmonary tuberculosis rely on a visualization technique where the area that contributes the most to categorization should also be the area that indicates TB. Gradient-based methods, such as the Grad-CAM, occlusion sensitivity, and the CRM, are used to visualize data. In addition, non-gradient techniques, such as the CAM, the Score-CAM, and attention maps, aid the visualization.

In conclusion, the promising results thus far suggest that it is possible to localize radiographic manifestations of pulmonary tuberculosis in a chest X-ray without pixel-level annotation. Furthermore, we have seen that providing a pixel-level annotation for a few data samples can improve weak localization results. As a result, weak localization reduces the cost of hiring a medical expert and the time required for data annotation. However, there is a decrease in detection performance compared to fully supervised semantic segmentation. Some techniques can only provide heatmaps, making it difficult to obtain accurate semantic segmentation. Even when using additional steps to obtain semantic segmentation, the accuracy is far below that of fully supervised semantic segmentation. In addition, weakly supervised segmentation algorithms might not be able to accurately identify regions of interest if the dataset is highly noisy and of low quality. Improving data quality through gathering high-quality data or employing pre-processing techniques can help to mitigate some of the challenges posed by weak localization algorithms. Finally, designing a robust and efficient algorithm is crucial in accurately localizing pulmonary tuberculosis manifestations in chest X-rays without pixel-level annotation. 

## Figures and Tables

**Figure 1 sensors-23-06781-f001:**
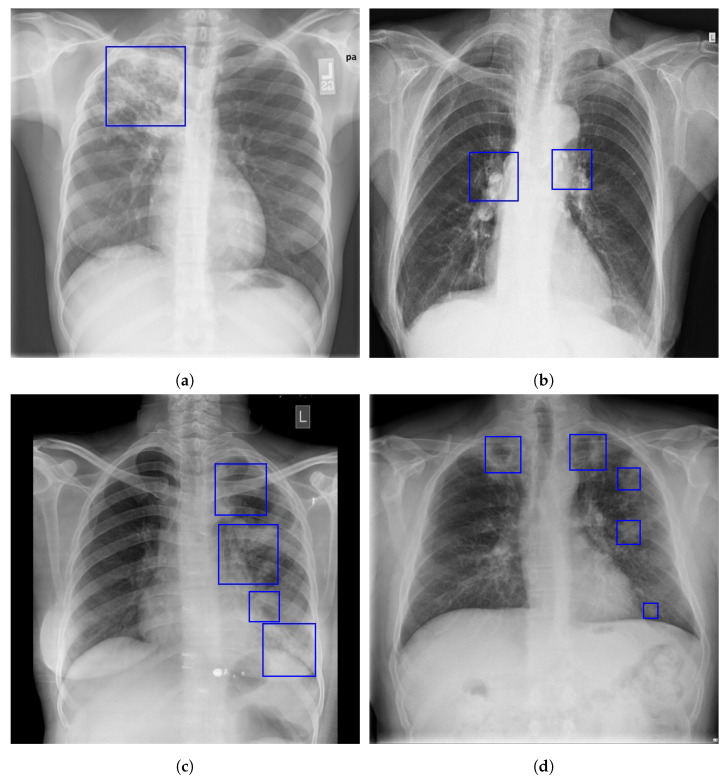
Examples of primary and post-primary PTB manifestation in a CXR [10]. The following are indicated by the blue boxes: (**a**) coalescing air space opacity on the right upper lobe; (**b**) enlarged hilar and mediastinal lymph nodes; (**c**) a thick-walled cavitary lesion in the left upper lobe; (**d**) bilateral apical thick-walled cavities and multifocal satellite air space opacities.

**Figure 2 sensors-23-06781-f002:**
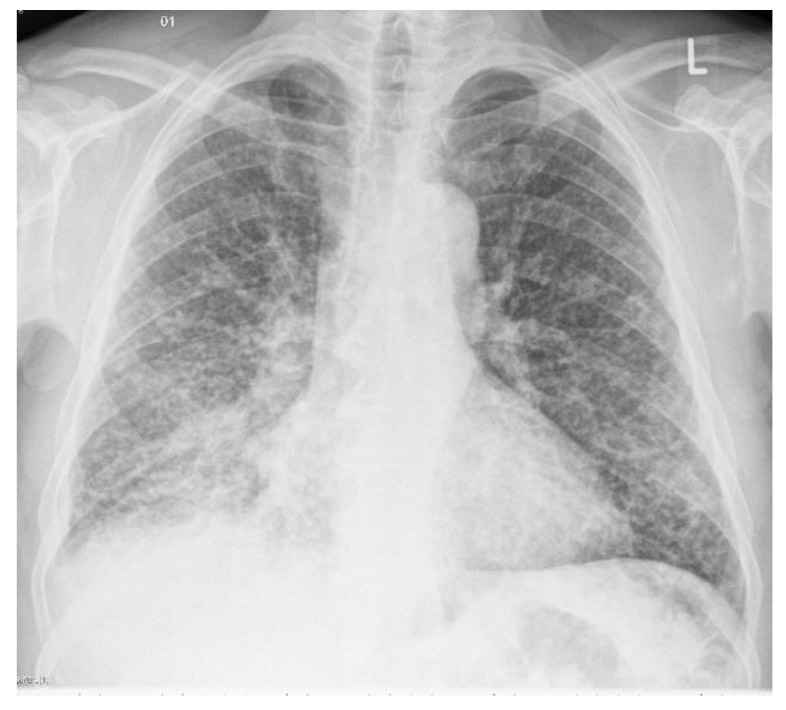
An example of chest X-ray images showing miliary tuberculosis with pulmonary nodules scattered throughout the lungs uniformly [10].

**Figure 3 sensors-23-06781-f003:**
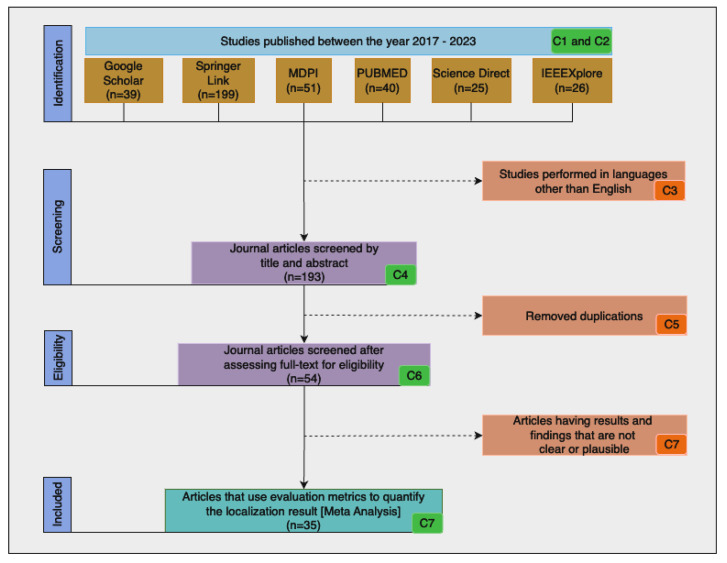
The flow diagram of paper selection.

**Figure 4 sensors-23-06781-f004:**
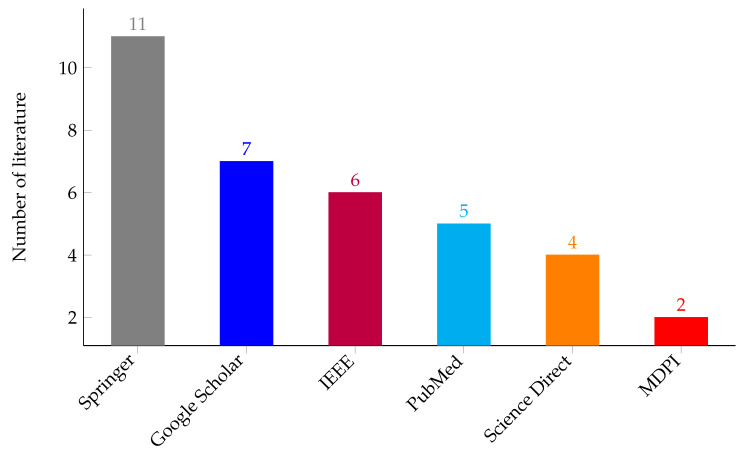
The number of research papers extracted from the search engines (Google Scholar, PubMed, MDPI, Science Direct, IEEE, and Springer).

**Figure 5 sensors-23-06781-f005:**
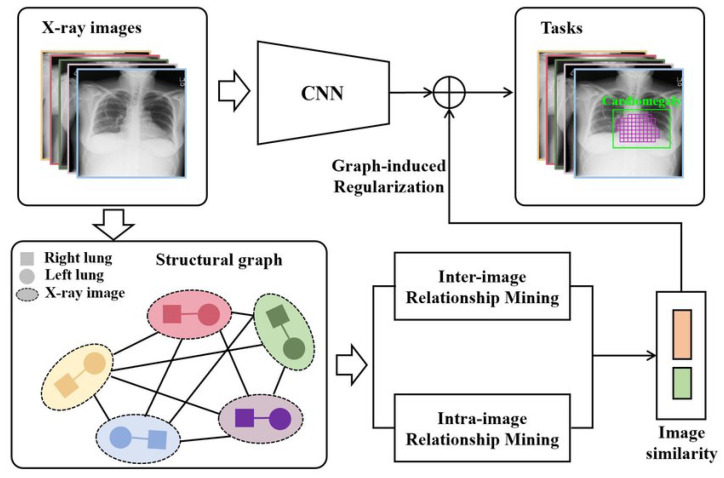
The general architecture of Graph Regularized Embedding Networks for weakly supervised disease localization in a chest X-ray image [64].

**Figure 6 sensors-23-06781-f006:**
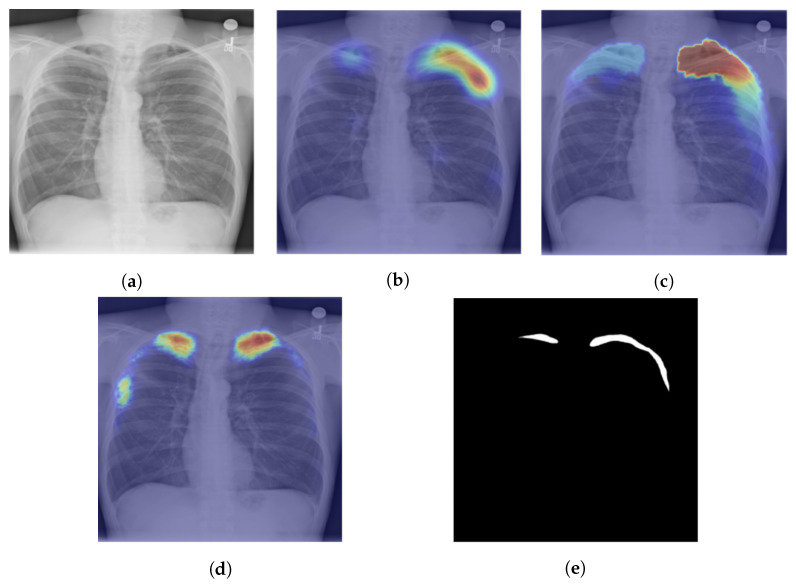
The three steps proposed in [89] for weakly supervised localization of tuberculosis manifestations in chest X-ray images with the red color representing areas where TB manifestation is highly likely: (**a**) chest X-ray image; (**b**) activation map generation with Grad-CAM++; (**c**) boundary improvements via IRNet; (**d**) segmentation; and (**e**) the ground truth where the pneumothorax manifested in the CXR.

**Figure 7 sensors-23-06781-f007:**
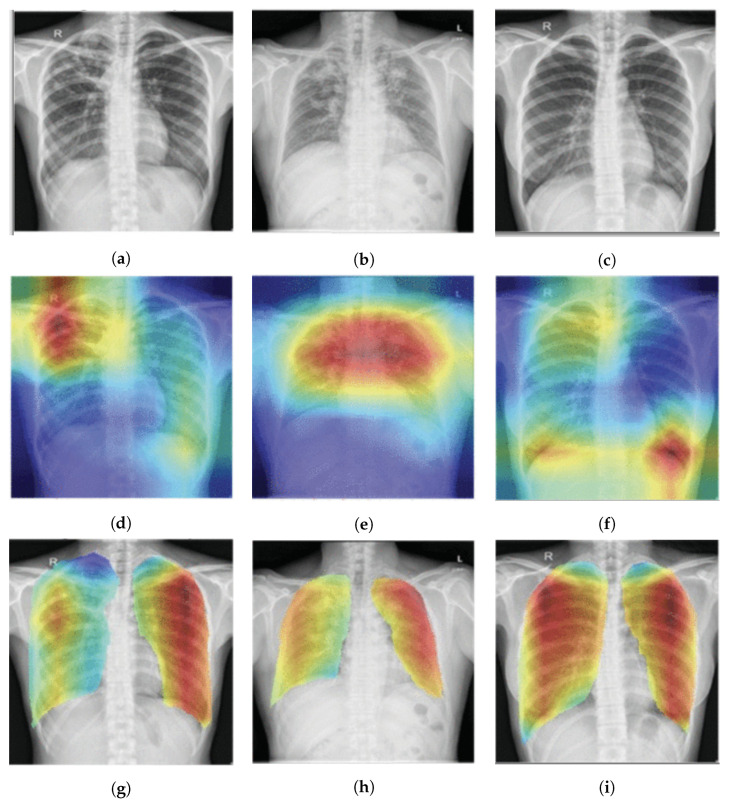
Score-CAM-based pulmonary TB localization [49] with the red color indicating areas where TB manifestation is highly likely: (**a**–**c**) show the original chest X-ray; (**d**–**f**) show heatmap results without the lung area segmentation; (**g**–**i**) show the heatmap result with lung area segmentation.

**Figure 8 sensors-23-06781-f008:**
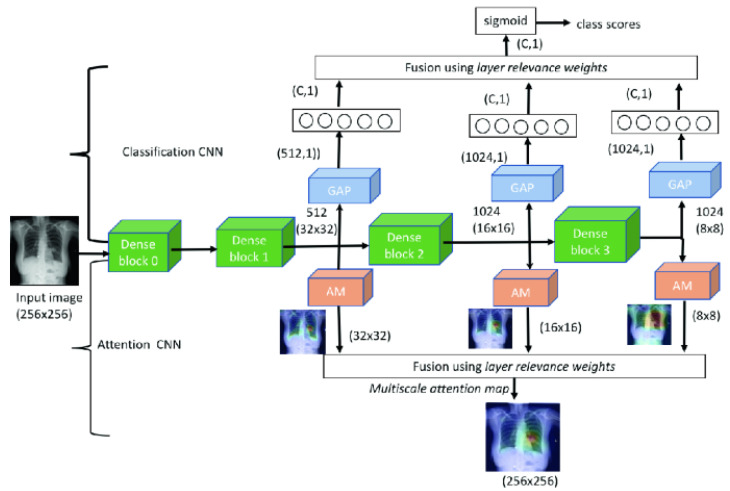
A framework proposed in [103] for localizing chest X-ray pathology.

**Figure 9 sensors-23-06781-f009:**
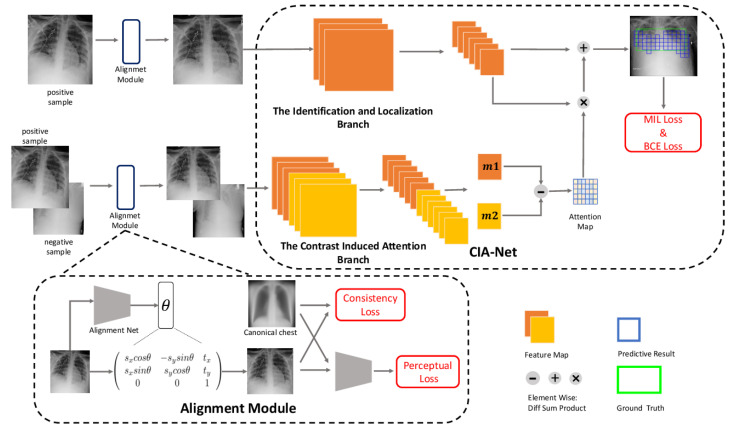
A contrast-induced attention network for chest X-ray diagnosis [105].

**Figure 10 sensors-23-06781-f010:**
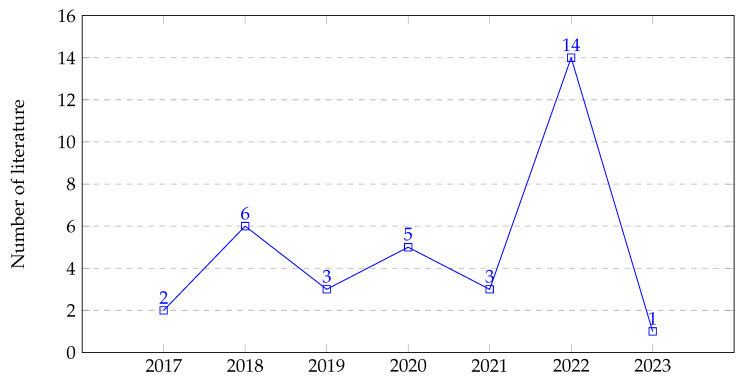
The number of research papers published from 2017 to 2023.

**Figure 11 sensors-23-06781-f011:**
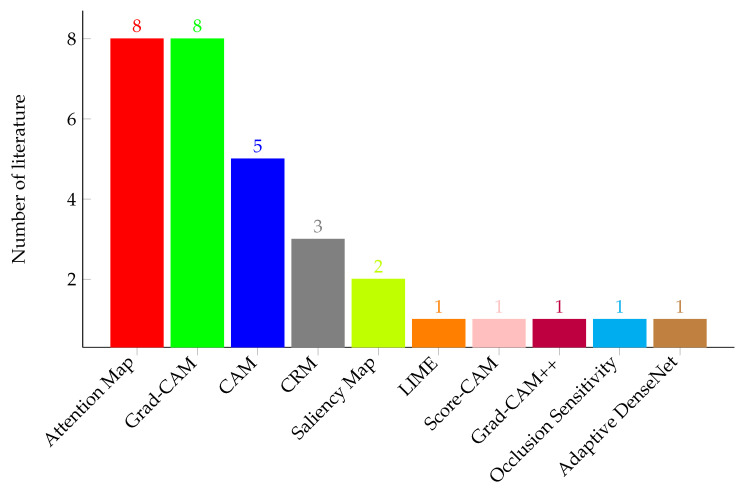
The number of research papers in terms of the techniques used for the weak localization of PTB in a chest X-ray image.

**Table 1 sensors-23-06781-t001:** Selection Criteria.

ID	Inclusion	Exclusion	Description
C1	X	-	Studies performed between the years 2017 and 2023
C2	X	-	Studies that are original article and conference papers
C3	-	X	Studies performed in languages other than English
C4	X	-	Studies that involve the localization and segmentation of pulmonary TB in a chest X-ray
C5	-	X	Duplicate publications
C6	-	X	Studies that use machine learning or DL methods
C7	-	X	Studies that are not presented with clear and plausible results
C8	X	-	Articles that use evaluation metrics to quantify the localization result

**Table 3 sensors-23-06781-t003:** Summary of the visualization techniques.

Approach	Description
Occlusion sensitivity	A technique used to understand the importance of different regions of an image in influencing the prediction of a deep neural network. It involves systematically, occluding different parts of the image and measuring the change in the prediction accuracy. By comparing the accuracy of the model with and without occlusion, we can determine the importance of different regions in the image.
Saliency maps	Saliency maps are heatmaps that highlight the regions of an image that are most important for the prediction made by a deep neural network. They are generated by computing the gradient of the prediction score with respect to the input image. Higher gradient values indicate regions that have a larger impact on the prediction, and these regions are visually highlighted in the saliency map.
Class activation map (CAM)	A technique used to generate a heatmap that highlights the discriminative regions of an image that contributed to a specific class prediction. It is typically used in convolutional neural networks (CNNs) with a global average pooling layer, and it involves multiplying the feature maps of the last convolutional layer with the weights of the fully connected layer corresponding to the predicted class.
Grad-CAM	The Grad-CAM is an extension of the CAM that overcomes some of its limitations. It generates a heatmap by taking the gradient of the predicted class score with respect to the feature maps of the last convolutional layer and then weights the feature maps based on the magnitude of the gradient. This allows the Grad-CAM to provide more fine-grained and localized explanations compared to the CAM.
Grad-CAM++	The Grad-CAM++ is an improved version of the Grad-CAM that further refines the localization accuracy of the heatmap by incorporating second-order gradients. It computes the second-order gradients of the predicted class score with respect to the feature maps, which provides additional information for determining the importance of different regions in the image.
Score-CAM	A technique that uses the predicted class score and the gradient of the class score with respect to the feature maps of the last convolutional layer to generate a heatmap. It weights the feature maps based on the product of the predicted class score and the gradient, which helps to highlight the regions that have a higher impact on the final prediction.
Class-selective relevance maps (CRMs)	A technique that generates relevance maps by combining the gradients of the predicted class score with respect to the input image and the gradients of the class score with respect to the feature maps. It uses a combination of global average pooling and global max pooling to capture both local and global information in the image.
Attention networks	A class of neural networks that dynamically focus on different regions of an image based on their importance for the task at hand. They use mechanisms such as self-attention or soft attention to assign weights to different regions of an image, which are then used to compute the final prediction. Attention networks are typically used in tasks that require sequential processing, such as machine translation or image captioning.

## Data Availability

Not applicable.

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
