# Peer review of "Weak Localization of Radiographic Manifestations in Pulmonary Tuberculosis from Chest X-ray: A Systematic Review"

_sensors, 2023, doi:10.3390/s23156781_

Round 1
Reviewer 1 Report
The article “Weak Localization of Radiographic Manifestations in Pulmonary Tuberculosis from Chest X-ray: A Systematic Review” is well-written and easy to read. My congratulations to the authors.
However, it presents some aspects that should deserve the authors' attention.
Regarding the abstract:
The objectives of the work need to be more evident in the summary, as well as the methodology used. Some results are presented, but the main conclusions of the work need to be presented. Authors must reformulate the abstract.
Line 2 (and in all document): Is written “radio-graphic” should be radiographic
Line: 11: Is written “deep learning” (DL)… should be Deep Learning (DL)
Figure 1: Should identify each image and a brief text description of what is seen in each image
Table 1 presents a summary of the related works. It would be better to submit it as a text with a greater description of its contributions and limitations. Presentation in tabular form is not an asset
Page7: Algorithm 4. Is a table or a figure?
Line 208: The section “Publicly Available Dataset for Tuberculosis Localization in Chest X-ray” and the following sections could already be the results of the work since it supports one of the research questions.
Tables 7 and 8 should be part of the Results section.
Conclusions Section
The second paragraph of the conclusions should be re-written, not mentioning acronyms (e.g. RQ3, RQ4…) but the summary on the findings according to the results.
Reviewer 2 Report
This research provides a review of Weak Localization of Radiographic Manifestations, and I have the following suggestions for further improvement:
1. The introduction section needs improvement by including a clear description of the problems addressed and highlighting the contribution of the research. This will help readers understand the context and significance of the study.
2. It would be beneficial to add dedicated sections on image enhancement techniques and normalization methods. These sections can explain the different approaches used to enhance the quality of radiographic images and normalize them for accurate localization.
3. Including sections on databases used in the research would be helpful. Additionally, the authors could consider referring to related review papers to gain insights and incorporate relevant ideas into their work.
1. Image segmentation for MR brain tumor detection using machine learning: A Review
TA Soomro, L Zheng, AJ Afifi, A Ali, S Soomro, M Yin… - IEEE Reviews in Biomedical Engineering, 2022.
2. Artificial intelligence (AI) for medical imaging to combat coronavirus disease (COVID-19): A detailed review with direction for future research
TA Soomro, L Zheng, AJ Afifi, A Ali, M Yin, J Gao - Artificial Intelligence Review, 2022.
Author Response
We would like to express our gratitude for the reviewer’s time and effort in providing feedback. We have taken the feedback seriously and have made an effort to address the issues raised point by point in the following manner.
- The introduction section needs improvement by including a clear description of the problems addressed and highlighting the contribution of the research. This will help readers understand the context and significance of the study.
- We have considered the reviewer’s comment and tried to highlight the contribution of the review on L124-119
- It would be beneficial to add dedicated sections on image enhancement techniques and normalization methods. These sections can explain the different approaches used to enhance the quality of radiographic images and normalize them for accurate localization.
- The purpose of this review is to identify the techniques used for the weak localization of tuberculosis in chest X-rays. However, we believe that adding section for image enhancement techniques and normalization methods would not directly contribute to the main topic.
- Including sections on databases used in the research would be helpful. Additionally, the authors could consider referring to related review papers to gain insights and incorporate relevant ideas into their work.
- Image segmentation for MR brain tumor detection using machine learning: A Review
TA Soomro, L Zheng, AJ Afifi, A Ali, S Soomro, M Yin… - IEEE Reviews in Biomedical Engineering, 2022.
- Incorporated into the related work as suggested by the reviewer on L108
- Artificial intelligence (AI) for medical imaging to combat coronavirus disease (COVID-19): A detailed review with direction for future research TA Soomro, L Zheng, AJ Afifi, A Ali, M Yin, J Gao - Artificial Intelligence Review, 2022.
- Incorporated into the related work as suggested by the reviewer on L105
Reviewer 3 Report
I think the paper is well written and thorough. The clinical problem is relevant but self-supervised/weakly supervised techniques are in general highly relevant to medical imaging problems.
Author Response
We appreciate the reviewer’s time and effort in providing feedback. However, we have decided not to take any action on this review report as the reviewer provided no comments.
Reviewer 4 Report
1. The authors should at least improve their figures and provide sufficient annotations related to the given discussions.
2. The discussions seem to focus more on the description of datasets rather than an actually reviewing them in-depth and their direct relevance to the actual topic.
3. The sections should have a better organization and flow, some sections seem to just be added without much reason why.
4. The authors should consider adding a nomenclature to provide better and easier way for readers to keep track of their variables and symbols.
5. The authors should provide more details regarding how they eliminated duplicate papers and how exactly they considered papers.
6. Some sections just don’t have enough coherence in the overall aim of the paper, the authors should improve this weakness.
7. Authors used algorithms to define their methods of search. However, the authors should at least clarify whether how they executed them, specifically in what manner.
8. The methodology behind the production of visual results, such as the Class Activation Maps (CAMs), remains unclear in the article. It is essential for the authors to provide a detailed explanation of how these visual results were obtained. Specifically, it is important to specify whether the authors sourced them from existing literature or conducted their own reproductions. Furthermore, to ensure the accuracy and reliability of these results, the authors should provide a clear evaluation of the fidelity of their reproductions in relation to the discussions and findings presented in the article. Including this information will enhance the transparency and credibility of their research.
9. The authors may also want to enhance the intuition behind their review.
The authors should consider investing additional effort into proofreading their manuscript to enhance the overall quality of the writing. This includes reviewing the text for grammar, punctuation, and spelling errors, as well as ensuring consistent and clear sentence structure. A well-polished manuscript will greatly improve the readability and professionalism of the research article.
Reviewer 5 Report
In this review article, the authors reviewed weak supervised learning techniques to locate radiographic manifestations of PTB. Even though I highly appreciate the motivation of this review and I feel that the topic is very interesting, it took me incredibly long time to review this paper, because this paper was substantially compromised by the writing. First, there are many grammar issues, showing that this paper was not carefully proofread. Second, all descriptions about “weakly supervised segmentation and localization” (specifically, Section 5) are extremely hard to understand. As a consequence, it is not helpful for the readers who want to learn relevant algorithms from this review. I think that the paper in the current form is not acceptable. It is essential to ensure the legibility, especially for such long review article.
I suggest that the authors should improve the writing of this paper thoroughly by carefully proofreading. In terms of the grammar, the authors may request the help of a native English speaker or editing service. More importantly, the corresponding author is responsible for the paper quality, such that all concepts, algorithms, and formulations should be clearly described. For revisions, I also have many detailed concerns and comments, which should be responded and addressed by the authors in a point-by-point manner.
1) Title: According to the PRISMA guideline, systematic review should include meta-analysis. As there is no meta-analysis, I feel that this is just a narrative review, instead of a systematic review.
2) L39: Please provide the full phrase of CXR.
3) L43: Full phrase of PPD?
4) L47, Section 1.2: The readers of this paper should include machine learning engineers who don't have much clinical experience regarding TB. As radiographic features are the key to diagnose TB as well as machine learning, I suggest that the authors should add a new figure. For each radiographic feature (such as cavitation, consolidation, mass, pleural effusion, calcification, and nodules), the authors should provide a subplot to visually present them, respectively. I think such visualization should be much helpful.
5) L49, “(TB)”: You don't need to provide the same abbreviation many times. Instead, you should make sure to define each abbreviation.
6) L72, Figure 1: What does the two subplots represent (left is normal, right is pathological)??? Please describe the two (left and right) sub-plots in the figure title, and mark Ghon lesions (e.g., using arrows) in Figure 1. In addition, if both subplots are pathological lungs, I suggest adding a normal lung radiograph for comparison.
7) L82, Figure 2: As you introduce a figure, you should carefully describe it to help the readers better understand your paper. Otherwise, if there is no specific purpose, there is no need to add a figure. In terms of Figures 2, I feel hard to differentiate it with Figure 1. Please expand the figure title to describe both subplots in Figure 2, and mark the disease areas that you attempted to describe in the subplots. Similarly, please improve all other figures!!!
8) L91: It is necessary to provide a figure to present miliary TB.
9) L191, section 3.2: According the PRISMA guideline, the authors should provide a flow diagram to present the results of literature screening (maybe in particular, in terms of weakly supervised learning).
10) L204, Table 7: The tables should be numbered in an increasing order. So Table 7 should be changed to Table 2 and moved here, and re-number other tables.
11) L279, Table 3: Where was Table 3 used in the manuscript?
12) L301-311: The description does not make sense. If predictions of a model are taken as pseudo-labels, then re-training of the model using the pseudo-labels will result in losses always equal to zero. Please improve your description about the self-training mechanism.
13) L331: What are super-pixels? Please define it.
14) L343, Figure 3: Once again, abbreviations should be defined prior to use. Please give the meanings of PAG and RAG!!!
15) L383: Eq. 6 (binary cross-entropy) missed a parenthesis.
16) L409: It is hard to understand this sentence. What do x_jj and y_ji mean by? Please clarify indices i and j.
17) L413: What is the meaning of "instead of only the classifier's parameters"???
18) L411-422: This paragraph is totally confusing. Please rephrase it.
19) L425: Where is the statement of Theorem 5.3?
20) L428: The symmetric function delta does not exist in Eq. 10???
21) L430-433: What is the relation of S(X) and P(X). Please further explain how to calculate P(X) from S(X)...
22) L434: Is Li et al's method related to Eq. 10? Please describe how to determine patch probability in Li et al's method.
23) L435, “PxP”: What is P? Please clarify...
24) L461: Meaning of "the classifier’s out decreases”???
25) L466-477: The paragraph about saliency map should be an individual section???
26) L470: (11) should be (Eq. 11)??? This is very confusing with reference citations. Also, please accurately express all other equations below. That is, add "Eq." before the numbers.
27) L473-475: This sentence is confusing. Please rephrase.
28) L484: “literary works” looks weird. Consider “previously reported works”.
29) L495: It is obvious that Eq. 15 describe how to calculate weights. Please double-check!!!
30) L504: In Eqs. 15 and 16, what is Fij_k???
31) L514, Figure 5: Where should Figure 5 be inserted in the manuscript? Also, what is the ground-truth label represent?
32) L528: X is image? Y is label? What are x0, x1, ..., xn???
33) L530, Eq. 19: What is the operation represented by a dot in the circle?
34) L531-536: Reading until now, I find that these descriptions have been totally not understandable!!! This cannot be accepted. First, please explain all letters in the equations 19 ~ 20. Then, rewrite the paragraph for clarity.
35) L537, Figure 6: What does the color in the heat map represent? For example, red means areas of diseased areas with high probability???
36) L540: Why "without the m_th hidden node"? Should be "with and without"?
37) L541: Where is the hidden node located? In a fully-connected layer?
38) L556: What does "GS" represent?
39) L573: Meaning of "calculate a pixel for the output"???
40) L585-597: What are positive and negative images? Also, the letters in Eq. 26 are completely not consistent with the description. Please provide detailed explanation on the computation of the weighted feature map. Furthermore, I think it is necessary to provide a flow chart to describe it.
41) L598: What is box-level annotation?
42) L598-607: Why use "we"? Did the authors develop this model?
43) L603: What is the meaning of samples here?
44) L612: What is GAP?
45) L634, Eq. 29: What are c and d?
46) L636: How to determine Loss_cls? Is this a weakly supervised learning approach?
47) L640-643: This sentence is too hard to understand. Please rephrase it to be clearer.
48) L644, “Discussion”: Please suggest which methods provide the state-of-the-art performance?
49) L649: Where were Figures 8-10 used in the manuscript???
50) L655: "the table in 7" should be "Table 7"
51) L657: Throughout the manuscript, it is not clear what the color values represent in a "heat map". Please briefly introduce heat maps in previous sections.
The writing needs to be improved thoroughly. Please see my comments above.
Round 2
Reviewer 4 Report
The authors have done an excellent job in addressing several of the mentioned concerns. However, there are still areas that need to be revised before considering it for publication.
Firstly, the abstract is excessively lengthy and should be condensed to provide a concise overview.
Additionally, the paper remains wordy and repetitive in certain sections, which should be revised to enhance clarity and conciseness.
The writing appears to be of good quality, but there is room for improvement in terms of making it more concise, yet still effective and accurate.
Author Response
Dear reviewer,
Thank you for your comments on our paper. We appreciate your time and effort in providing feedback. We have carefully considered your comments and we have made several changes to the paper in response. We hope that you will consider our response.
Thank you for your feedback
Firstly, the abstract is excessively lengthy and should be condensed to provide a concise overview.
- We have rewritten the abstract and made it shorter as suggested by the reviewer.
Additionally, the paper remains wordy and repetitive in certain sections, which should be revised to enhance clarity and conciseness.
- Additional proofreading has been made to avoid repetition and enhance clarity.
Comments on the Quality of English Language
The writing appears to be of good quality, but there is room for improvement in terms of making it more concise, yet still effective and accurate.
- To address this issue, we proofread and improved the English quality.
Reviewer 5 Report
Although the paper has been improved somehow by partially responding to my comments, it appears that many previous comments have been ignored. In particular, I feel surprised that the authors deal with many comments in a rough way, considering the corresponding author from the well-known University of Ulm, who may not provide any guidance for writing a scientific achieved journal article.
The authors should be aware that the reviewers are providing free service and do the best to help improve the paper quality. Correspondingly, the authors should respect each comment of the reviewers. I really think that the authors should respond to the reviewers in a proactive attitude and are truly willing to address the reviewers' concerns.
There still are many grammar issues and unclear descriptions, especially for the section to describe the weakly supervised learning algorithms. These problems have severely impeded the understanding of your work. As a reviewer, I am not responsible for correcting each issue in the writing, which required the authors to carefully proofread.
In addition, for the convenience of the authors, I provided detailed locations which need to revise for the authors. I don't know why the authors did not provide line numbers in their response to conveniently position the revisions for the reviewers.
Below are a list of my comments that were not well addressed:
1) Title: According to the PRISMA guideline, systematic review should include meta-analysis. As there is no meta-analysis, I feel that this is just a narrative review, instead of a systematic review. - The meta-analysis has been provided in Figure 4 along with the flow diagram.
It appears that the authors are not familiar with the concept of meta-analysis. Meta-analysis is statistical analysis of pooled reported data. Through this paper, I didn’t find any meta-analysis performed by the authors.
4) L47, Section 1.2: The readers of this paper should include machine learning engineers who don't have much clinical experience regarding TB. As radiographic features are the key to diagnose TB as well as machine learning, I suggest that the authors should add a new figure. For each radiographic feature (such as cavitation, consolidation, mass, pleural effusion, calcification, and nodules), the authors should provide a subplot to visually present them, respectively. I think such visualization should be much helpful.
Where is the revision for the comment? If you don't want to change, please give your reasons.
12) L301-311: The description does not make sense. If predictions of a model are taken as pseudo-labels, then re-training of the model using the pseudo-labels will result in losses always equal to zero. Please improve your description about the self-training mechanism. - This technique works under a certain presumption: the dataset should adhere to the presumption that two examples belonging to the same class can be connected by a series of edges in a graph.
Where is the improved description about the self-training mechanism?
13) L331: What are super-pixels? Please define it. - Superpixels are the result of the perceptual grouping of pixels, and it is defined in the reference provided on L340.
I have downloaded the reference [56] cited at L340, and I did not find “superpixels” occurring this paper.
14) L343, Figure 3: Once again, abbreviations should be defined prior to use. Please give the meanings of PAG and RAG!!! - Addressed by replacing the abbreviation with its full phrase.
Can you show me the location of these full phrases?
15) L383: Eq. 6 (binary cross-entropy) missed a parenthesis.
- Addressed by providing the missed parenthesis.
The authors did not correct the equation!
16) L409: It is hard to understand this sentence. What do x_jj and y_ji mean by? Please clarify indices i and j. - x_ij denotes the pixel at the i,j elements of the image matrices. The sentence is rewritten to make it clear.
You changed y_ji to y_k. What is the subscript k?
18) L411-422: This paragraph is totally confusing. Please rephrase it.
- Paraphrase the paragraph to make it clear and unambiguous.
There is not any improvement for this paragraph.
19) L425: Where is the statement of Theorem 5.3? - Theorem 5.3 and Eq.10 are found to be confusing and have no significance in explaining the concept. As a result, we drop both the theorem and the Equation and explained the concept of MIL in a clear and understandable way.
20) L428: The symmetric function delta does not exist in Eq. 10??? - Theorem 5.3 and Eq.10 are confusing and have no significance in explaining the concept. As a result, we drop both the theorem and the Equation and explained the concept of MIL in a clear and understandable way.
21) L430-433: What is the relation of S(X) and P(X). Please further explain how to calculate P(X) from S(X)... - Theorem 5.3 and Eq.10 are confusing and have no significance in explaining the concept. As a result, we drop both the theorem and the Equation and explained the concept of MIL in a clear and understandable way.
For comments 19-21, I am surprised that the authors address the reviewer's comments by removing the paragraphs that were not clearly described.
31) L514, Figure 5: Where should Figure 5 be inserted in the manuscript? Also, what is the ground-truth label represent? - Figure 5 which is now Figure 6 has been inserted in the manuscript on L517,
You did not explain the meaning of the ground-truth label in Fig. 6.
35) L537, Figure 6: What does the color in the heat map represent? For example, red means areas of diseased areas with high probability??? - Yes, the red color indicates the region that highly contributes to the classification, hence it is the one that has a disease pattern.
Why not add the explanation in the figure title?
38) L556: What does "GS" represent? - It is Generative Synthesis. Its full phrase is added to the manuscript now.
I did not find the full phrase provided by the authors.
42) L598-607: Why use "we"? Did the authors develop this model? - Addressed by removing the pronoun and rewriting the sentence.
You did not completely address it.
47) L640-643: This sentence is too hard to understand. Please rephrase it to be clearer. - The sentence is rewritten to make it clear and understandable as suggested by the reviewer.
I don't think the authors have made any changes.
48) L644, “Discussion”: Please suggest which methods provide the state-of-the-art performance? - Addressed by suggesting studies that provided robust methods with clear evaluation metrics o L664-668.
Please explicitly described the names of the methods that you suggested. For example, you may say something like: “According to our review, we found that Attention Map and Grad CAM achieve the best performance in terms of contemporary implementations”.
It is extremely necessary to thoroughly improve the writing, which has severely impede the understanding of this work.
